# Study on Low-Temperature Index and Improvement Measures of Emulsified Asphalt Cold Recycled Mixture

**DOI:** 10.3390/ma15217867

**Published:** 2022-11-07

**Authors:** Xiaoqing Yu, Zhanchuang Han, Yu Cai, Liping Liu, Lijun Sun

**Affiliations:** 1Jiangxi Ganyue Expressway Co., Ltd., Nanchang 330009, China; 2The Key Laboratory of Road and Traffic Engineering, Ministry of Education, Tongji University, No. 4800 Cao’an Rd., Shanghai 201804, China

**Keywords:** cold recycled mixture, low-temperature performance, low-temperature index, fiber, fracture work, fracture energy

## Abstract

With the promotion of cold recycled mixture (CRM) in cold areas and the improvement of its application layer, the enhancement of the low-temperature performance of mixtures is particularly important. The applicability of the current low-temperature bending test method to CRM is controversial. Firstly, the low-temperature crack resistance of CRM with different gradations and emulsified asphalt contents was studied by the indirect tensile (IDT) test and the semi-circular bending (SCB) test. Thereafter, the low-temperature performance evaluation index suitable for CRM was put forward. Then, the low-temperature performance of CRM with different gradations, fiber types, and contents was evaluated by using the above low-temperature evaluation index. The results show that the low-temperature performance of CRM with different gradations and emulsified asphalt contents can be distinguished by fracture work (W) and fracture energy (G_f_). Not only do the test results have little variability (about 12% and 15%, respectively), but also the correlation coefficient with the new asphalt film thickness is the highest (0.8595 and 0.8939, respectively). Compared with coarse gradation (AC-25) and fine gradation (AC-13), medium-gradation (AC-20) CRM has higher low-temperature performance, and polyester fiber can significantly improve the low-temperature performance of CRM. Compared with non-fiber, the W and G_f_ of CRM of polyester fiber (0.3% content) can be increased by at least 42% and 30%, respectively.

## 1. Introduction

CRM is a kind of green low-carbon, energy-saving, and environmental protection renewable material. Due to the higher utilization rate of old materials, lower energy consumption, and environmental pollution, CRM has been widely used in the reconstruction and expansion of asphalt pavement in China in recent years [1,2,3,4]. Nevertheless, the cold weather in the north of China will have a negative influence on the application of CRM. Therefore, it is necessary to search for methods to improve the low-temperature crack resistance of CRM. Many scholars have conducted a lot of work to improve the low-temperature crack resistance of CRM. Therefore, the main improvement measures are using modified emulsified asphalt, changing the amount of new aggregate, adding various additives, composite modification of modified emulsified asphalt and regenerant, etc. Representative scholars include Moon, Ki Hoon, Pasetto, Marco, Dolzycki, Bohdan, Liu, Wang, Hao, Qin, etc. [5,6,7,8,9,10,11]. The above measures play a positive role in improving the low-temperature performance of CRM. In addition to the above measures, the fiber in asphalt mixtures has the function of reinforcing support, dispersing and transferring stress, and limiting the relative slip of mineral aggregate. It has achieved good results in slowing down the fatigue crack growth of mixtures and can be used to improve the low-temperature crack resistance of CRM [12,13]. In addition, coarse- and fine-aggregate gradation also have an important impact on the performance of asphalt pavement. In particular, when CRM is used in cold regions, the selection of a reasonable gradation should focus on the low-temperature crack resistance [14,15].

However, before commencing research on the improvement measures of low-temperature performance, an appropriate low-temperature performance evaluation index should be found. At present, there are three kinds of low-temperature evaluation indexes for asphalt mixtures: the first is energy evaluation indicators, mainly including fracture work, bending strain energy, fracture energy, etc.; the second is the energy-normalized index, mainly including fracture toughness, JC, crack resistance index, toughness index, etc.; and the third is the tolerance value of fracture strength and fracture strain, mainly including splitting strength, bending tensile strength, stress intensity factor, fracture strength index, maximum displacement, maximum bending tensile strain, etc. [16]. Many scholars have made a lot of contributions to the research on low-temperature test methods and evaluation indexes. In the 1990 Strategic Highway Research Plan, the United States developed a temperature stress test of constrained specimens and took JC as the index to evaluate the low-temperature performance of asphalt mixtures [17,18]. However, the disadvantage of this method is that it does not consider the evolution of cracks over time and the impact of load on them. Therefore, the IDT test was developed to evaluate the low-temperature performance of asphalt mixtures [19,20]. In the research of the “Seventh Five-Year Plan” and the “Eighth Five-Year Plan”, China carried out a series of tests on the low-temperature performance of asphalt mixtures, including stress relaxation testing; low-frequency, low-temperature, repeated-load testing; bending testing; splitting testing; and creep testing, mainly evaluating the low-temperature performance of asphalt mixtures with mechanical indexes [21,22,23]. Representative scholars include Zhang, Zhu, Feng, Wang, Huang, etc. [24,25,26,27,28,29]. The above evaluation indexes have their own advantages, and no unified standard has been formed at present. In the Chinese asphalt pavement construction technical specification, the maximum bending tensile strain of low-temperature bending specimens is used as the evaluation index of low-temperature performance for asphalt mixtures, and good engineering application effects have been achieved. However, considering the characteristics of high void ratio, poor uniformity, and cohesiveness of CRM, the surface particles of the bending specimen prepared are loose and easy to fall off, and there is a common phenomenon of missing corners around, so the bending tensile strain test results have a great variability [4,11]. If the low-temperature bending test is used to evaluate the low-temperature performance of CRM, it will lead to inaccurate evaluation results and affect the engineering application effect [11,26]. How to select a reasonable low-temperature evaluation index of CRM has not been effectively solved, and the selection of the low-temperature index needs further experimental demonstration.

Based on the above issues, in this paper, the low-temperature evaluation indexes and performance improvement measures of CRM are studied. First, the SCB test and the IDT test are used to evaluate the low-temperature performance of CRM under different test schemes. By analyzing different low-temperature evaluation indexes, the low-temperature evaluation indexes suitable for CRM are proposed. Then, the evaluation index is used to analyze the low-temperature performance of CRM under different gradations, fiber types, and contents. The low-temperature evaluation index and low-temperature performance improvement method of CRM suitable for cold regions are recommended. The research results can be used as a reference for cold recycled projects in the field.

## 2. Raw Materials and Experimental Design

### 2.1. Raw Materials

(1)Emulsified asphalt

The specific indices of the emulsified asphalt used in this test are listed in Table 1.

(2)RAP and Crushed Stone

RAP means reclaimed asphalt pavement material. In this paper, the source of old materials is the old asphalt pavement milling material of high-speed reconstruction and expansion, which is classified into the following three grades: 0–5 mm, 5–10 mm, and 10–30 mm. The crushed stone is limestone, and its grade is 10–20 mm.

(3)Mineral Powder, Cement, and Water

Mineral powder and cement were acquired from the local producer in Inner Mongolia, China. The technical properties are shown in Table 2 and Table 3. Their technical indices met the technical requirements specified in the technical specification for the construction of Highway Asphalt Pavement (JTG F40-2004); tap water was used as the mixing water.

(4)Fiber

The technical indexes of polyester fiber, polypropylene fiber, basalt fiber, and lignin fiber are shown in Table 4.

### 2.2. Low-Temperature Test Method

(1)IDT Test

The IDT test is a test method developed by the Texas Transportation Research Center for evaluating the crack resistance of asphalt mixtures [30]. When it is used to evaluate the crack resistance of asphalt mixtures at low temperatures, a temperature of −10 °C ± 0.5 °C and a loading rate of 1 mm/min should be adopted. For the IDT test, crack resistance index (*I*_CT_) and fracture work are the main evaluation indicators. *I*_CT_ reflects the anti-crack propagation performance of asphalt mixtures. It is reported that the higher the crack resistance *I*_CT_ is, the better the low-temperature performance of the mixture is. *I*_CT_ can be calculated by Equation (1).
(1)$$I_{\mathrm{CT}}=\frac{t\times G_{f}\times L_{75}}{62\times \left|m_{75}\right|\times D}$$
where t is the thickness of the specimen, mm; D is the diameter of the specimen, mm; |m_75_| is the specimen load when it falls to 75% by the peak maximum load and absolute slope of the load–displacement curve; L_75_ is the deformation of the specimen when the load drops from the peak value to 75%, mm; G_f_ is the fracture energy of the specimen, N/m.

The fracture work, the area enclosed by the first half of the peak value curve on the force–displacement curve, is used to characterize the anti-crack initiation performance of asphalt mixtures. At the same time, the higher W can offer the stronger cracking resistance of asphalt mixtures at low temperatures, as listed in Equation (2).
(2)$$W=\;\int_{0}^{L}Pdl$$
where *L* is the deformation of the specimen when the load reaches the maximum, mm; *P* is the maximum load force of the specimen, kN.

(2)Semi-Circular Bending Test

The semi-circular bending test method, referred to as the SCB test, is a method based on the principle of fracture mechanics. It was originally used to analyze the fatigue cracking of hot mix asphalt. Then, the SCB test was later introduced into the American AASHTO test specification to evaluate the low-temperature cracking resistance of asphalt mixtures [31]. The corresponding evaluation indicators are composed of fracture energy and fracture toughness (K_IC_). G_f_ refers to the work done by fracture expansion of the per unit area, that is, the external work absorbed per unit area of fracture. It is generally believed that the higher G_f_ is, the stronger resistance cracking of asphalt mixture to low temperatures is. G_f_ can be calculated by Equation (3).
(3)$$G_{f}=\frac{W_{f}}{A_{\mathrm{lig}}}$$
where W_f_ is the integral of specimen load force and displacement, A_lig_ is the ligament area, A_lig_ = (R − A) × t, r is the specimen radius, A is incision length, and t is specimen thickness.

K_IC_ can also represent the low temperature performance of asphalt mixtures, which can reflect the energy absorption capacity of the material in the fracture process. The greater the toughness is, the stronger the ability of the specimen to prevent crack propagation is. K_IC_ can be calculated by Equations (4) and (5).
(4)$$K_{IC}=\frac{P_{\max}({\prod a)}^{1/2}}{2\times \mathrm{BR}\times Y_{I}\left(0.8\right)}$$
(5)YI0.8=4.872+1.219×(aR)+0.063×e7.045(a/R)
where P_max_ is the peak load, Y_I_(0.8) is the standardized stress intensity factor, R is the specimen radius, and a is the incision length.

J_C_ is used to evaluate the fracture performance of cracked asphalt mixtures based on the fracture mechanics theory. Relevant studies show that J_C_ has a good correlation with the crack rate of asphalt pavements. Generally, it is believed that the higher J_C_ is, the better the low-temperature performance of asphalt mixtures is. J_C_ can be calculated by Equation (6).
J_c_ = (U_1_/B_1_ − U_2_/B_2_) × (1/(a_1_ − a_2_))(6)
where J_C_ is the fracture toughness (kJ·m^−2^), U is the strain energy (kJ), it is the load–displacement curve area from the beginning of load application to the peak value, *a* is the crack depth, and *B* is the specimen thickness.

### 2.3. Specimen Forming

The Marshall specimens’ forming method based on the principle of secondary thermal compaction (earlier proposed by our research group) was adopted; that is, after 100 times of compaction at room temperature, each specimen was placed on the ground for 24 h at room temperature, and then it was placed into the oven at 60 °C for 48 h. After taking it out, it was compacted 50 times, cooled to room temperature (generally, for 6 h), the mold was removed, and the specimen was prepared. The rule of “150 + 75” compaction times was adopted for large Marshall specimens, from which the SCB specimens were cut (with a diameter of 150 mm, a height of 75 mm, and a thickness of 45 mm). A notch with a width of 2.5 mm and a depth of 15 mm was cut in the middle of each SCB specimen. The cutting method was recommended in the technical specification for cold recycling construction of emulsified asphalt on Highway Asphalt Pavement (DB13/T-2020) [11,32,33].

### 2.4. Test Scheme

(1)Study on Low-Temperature Crack Resistance Index of CRM

By controlling the passing percentage of the 4.75 mm sieve, the gradation of CRM was adjusted. Accordingly, the proportion of aggregates in each grade was changed, triggering the passing percentage of the 4.5 mm sieve as 35%, 40%, 45%, and 50%, respectively. The passing percentage of each sieve is shown in Figure 1. For the above four gradations, the amount of emulsified asphalt of 3.3%, 3.6%, and 3.9% was used for each gradation. The optimum water content of the four gradations was determined by the geotechnical method, reaching 5.2%, 5.1%, 4.9%, and 4.8%, respectively.

The IDT test and the SCB test were carried out with the 12 types of CRM, respectively. Afterwards, low-temperature evaluation indexes such as maximum peak force (P_100_), maximum displacement (Ls), |m_75_|, material stiffness (S), *I*_CT_, fracture work, K_IC_, and fracture energy were calculated. Then, the various evaluation indexes on the low-temperature crack resistance of CRM under different working conditions were compared and analyzed.

(2)Study on Low-Temperature Performance of CRM with Different Gradation Types

Three CRM gradations (represented by Gradations 5–7) of fine-grained (AC-13), medium-grained (AC-20), and coarse-grained (AC-25) were designed, respectively. The passing rate of each sieve is shown in Table 5. The three grading curves are shown in Figure 2. The low-temperature performance index selected in (1) was used to evaluate the low-temperature performance of the above three types of CRM with coarse, medium, and fine grading. For the above three gradations, the optimum water content determined by the geotechnical method is 5.7%, 5.2%, and 4.9%, respectively. According to the cold recycling technical specifications, the optimum emulsified asphalt dosage of the above three gradations is determined to be 3.0%, 3.5%, and 3.9%.

(3)Influence of Fiber Type and Content on Low-Temperature Performance of CRM

Polyester fiber (PF), basalt fiber (BF), lignin fiber (LF), and polypropylene fiber (PPF) were mixed into CRM, respectively, with an amount of 0.3% [34,35]. Taking the dry splitting strength as an index, the optimal water consumption of CRM under each fiber is determined to be 5.8%, 5.5%, 6.2%, and 5.8%, respectively. Then, the optimized low-temperature performance index in (1) was used to evaluate the impact of different fiber types on the low-temperature performance of CRM. After selecting the best fiber type, the fiber content was changed to 0%, 0.1%, 0.2%, 0.3%, and 0.4%. The low-temperature performance of CRM with different fiber contents was tested, and then the optimal fiber content was determined.

## 3. Test Results and Analysis

### 3.1. Comparison and Analysis of Low-Temperature Index of CRM

(1)Low-temperature index of SCB test

The test results of fracture energy, S, K_IC_, and J_C_ of CRM under different gradations and emulsified asphalt contents are shown in Figure 3.

It can be seen from Figure 3 that when the gradation is unchanged, the fracture energy and fracture toughness of CRM increase with the increase in emulsified asphalt content. When the emulsified asphalt content is unchanged, the coarser the gradation, the smaller the fracture energy and fracture toughness of CRM. This is because with the increase in emulsified asphalt content, the asphalt film thickness per unit area of aggregates is increased so that the aggregates are more fully bound, and the bonding strength between aggregates is increased, thus increasing the low-temperature crack resistance of CRM. However, when the amount of emulsified asphalt is unchanged, the coarser the gradation, the harder it is for asphalt or asphalt mortar to completely wrap the coarse aggregate in the aggregate. In the process of crack propagation, this part cannot form an effective bond and thus shows poor low-temperature crack resistance. However, with the change of gradation and emulsified asphalt content, the stiffness and J_C_ of CRM do not show regular changes. This shows that fracture energy and fracture toughness can better distinguish the low-temperature performance of CRM with different gradations and emulsified asphalt contents, but stiffness and J_C_ cannot be well discriminated. As previously mentioned, this is a common and convicted conclusion that the coarser the gradation is, the easier the mixture is, resulting in the characteristics of brittleness under low-temperature environments [36]. Simultaneously, from the variability of test results, the variability of fracture energy data is about 10%, and the variability of fracture toughness data is more than 20%. Therefore, when the SCB test is used to evaluate the low-temperature performance of CRM, the fracture energy index is recommended.

(2)Low-Temperature Index of IDT Test

The test results of Ls, |m_75_|, *I*_CT_, and fracture work of CRM under different gradations and emulsified asphalt dosages are shown in Figure 4.

It can be seen from Figure 4 that when the gradation is unchanged, the fracture work and the anti-cracking index of CRM increase with the increase in the emulsified asphalt content. When the emulsified asphalt content is unchanged, the coarser the gradation, the smaller the fracture work and the anti-cracking index of CRM. This is consistent with the low-temperature performance test results in Figure 3. For example, when Gradation 1 is used, the fracture work of CRM at the dosage of 3.3%, 3.6%, and 3.9% emulsified asphalt is 14.9 N/m^−1^, 16.3 N/m^−1^, 18.4 N/m^−1^, respectively, and the crack resistance index is 28.84, 45.49 and 68.92, respectively; When the emulsified asphalt content is 3.3%, the fracture work of CRM of Gradings 1–4 is 14.9 N/m^−1^, 13.1 N/m^−1^, 11.5 N/m^−1^, and 8.7 N/m^−1^, respectively, and the anti-cracking index is 28.84, 24.76, 20.77, and 11.96. However, with the change of mineral aggregate gradation and emulsified asphalt content, the Ls and |m_75_| of CRM do not show regular changes. From the above test results, it can be seen that the fracture work and the anti-cracking index of the IDT test can well distinguish the low-temperature performance of CRM under different gradations and emulsified asphalt dosages. From the variability of test results, the variability of the fracture work results is about 10%, and the variability of the crack resistance index results is more than 25%. Therefore, when the IDT test is used to evaluate the low-temperature performance of CRM, the fracture work index is recommended.

(3)Correlation Analysis

Relevant studies have shown that under a certain amount of asphalt, the asphalt film thickness has a positive correlation with the low-temperature performance of the asphalt mixture [36,37,38]. This paper analyzes the correlation between the new asphalt film thickness of CRM and the low-temperature performance test results of CRM under different gradations and emulsified asphalt dosages. The new asphalt film thickness refers to the asphalt film thickness wrapped on the RAP surface after the demulsification of newly added emulsified asphalt. The results of the above analysis are used to evaluate the applicability of each index to the low-temperature performance of CRM. The correlation analysis results are shown in Figure 5.

Among them, refer to technical specifications for construction of Highway Asphalt Pavement (JTG F40-2004) [39], according to the formula DA = P_be_ × 10/(γ_b_×SA), calculate the new asphalt film thickness of CRM under different conditions, where DA is the effective thickness of asphalt film, P_be_ is the effective asphalt content, γ_b_ is the relative density of asphalt (25 °C), and SA is the specific surface area of aggregate. The calculation results are shown in Table 6.

It can be seen from the correlation analysis results in Figure 5 that with the increase in the thickness of the new asphalt film, the corresponding *I*_CT_, fracture work, fracture energy, K_IC_, and J_C_ gradually increase, |m_75_| gradually decreases, and Ls also gradually increases, indicating that CRM becomes soft, the deformation resistance increases, and the crack growth rate also decreases. Therefore, the low-temperature performance of CRM is improved. The change rule of S is contrary to that of other indexes, which indicates that this index is not suitable for evaluating the low-temperature performance of CRM. At the same time, the correlation coefficient results show that the fracture work, fracture energy, and K_IC_ are all greater than 0.8, and the other indicators are all lower than 0.7, which indicates that the above three indicators are sensitive to the change of new asphalt film thickness. Therefore, considering the discrimination of each indicator to different test schemes and the variability of data results, this paper suggests that when evaluating the low-temperature performance of CRM, for the SCB test method, The fracture energy index is recommended, and the fracture work index is recommended for the IDT test method.

### 3.2. Effect of Different Gradations on Low-Temperature Performance of CRM

The test results of voids, dry splitting strength (R_T_), fracture work and fracture energy, dry wet splitting strength ratio (DWR), freeze–thaw splitting strength ratio (TSR), dynamic stability (DS), and unconfined compressive strength (UCS) of fine-grained (AC-13), medium-grained (AC-20), and coarse-grained (AC-25) CRM are shown in Figure 6. CRM with different gradation types.

It can be seen from Figure 6 that the voids and dry splitting strength (R_T_) fine-grained (AC-13), medium-grained (AC-20), and coarse-grained (AC-25) CRM gradually increase, which indicates that the coarser the gradation, the greater the dry splitting strength, but the final three gradations meet the road performance requirements. Among the coarse, medium, and fine gradations, the fracture work and the fracture energy of the medium-grained (AC-20) CRM are the largest, which are at least 32% and 24% higher than those of the fine-grained (AC-13) CRM, and at least 52% and 41% higher than those of the coarse-grained (AC-25) CRM. Jiang et al., Cai et al., and others also believe that the medium-grain (AC-20) graded CRM has better low-temperature performance [34,35]. Therefore, when CRM is applied to the surface structure, considering the low-temperature crack resistance of the pavement, medium-grained (AC-20) CRM is recommended.

### 3.3. Effect of Different Fiber Types and Dosages on Low-Temperature Performance of CRM

(1)Optimization of fiber type

The test results of voids, dry splitting strength (R_T_), fracture work, and fracture energy of CRM of polyester fiber (PF), basalt fiber (BF), lignin fiber (LF), polypropylene fiber (PPF), and no fiber (NF) are shown in Figure 7.

It can be seen from Figure 7 that among polyester fiber, basalt fiber, lignin fiber, and polypropylene fiber, CRM with polyester fiber has the largest dry splitting strength (R_T_), fracture work and fracture energy. Compared with CRM without fiber, the dry splitting strength (R_T_), fracture work, and fracture energy are increased by at least 16%, 73%, and 30%. In previous studies, Wang et al., Hao et al., and others also believed that polyester fiber had a more obvious effect on improving the low-temperature performance of CRM [40,41]. Therefore, polyester fiber is recommended to be used. Subsequently, the optimal content of polyester fiber was optimized.

(2)Optimization of fiber content

The test results of voids, dry splitting strength (R_T_), fracture work, and fracture energy of CRM under different fiber contents are shown in Figure 8.

It can be seen from Figure 8 that with the increase in polyester fiber content, the fracture work and the fracture energy of CRM increase at first and then gradually decrease. This is because when the content of fiber is small, the dispersion of fiber in the mixture is better, which can play the role of reinforcement and consolidation and constraint of crack growth, so the low-temperature crack resistance of CRM is improved. However, when the fiber content exceeds an appropriate amount, the dispersion of the fiber in the mixture will become poor. Not only can it not play a reinforcing and stabilizing role, but it will become a mechanical weak point for the crack propagation of CRM, which will cause the large particle aggregate to be squeezed away, and the cohesion between the slurry and the aggregate will become smaller, resulting in the decrease in the fracture work and the fracture energy. When the content of polyester fiber is 0.3%, the fracture work and the fracture energy of CRM are both the largest, and the fracture work and the fracture energy of CRM are at least 73% and 30% higher than those of CRM without fiber. Therefore, considering the low-temperature crack resistance of CRM, a fiber content of 0.3% is recommended.

## 4. Conclusions

Low-temperature crack resistance is one of the important properties of CRM, which directly affects the promotion and application of the cold recycled layer in the upper layer of the cold region. In this paper, different low-temperature evaluation indexes are analyzed, the measures to improve the low-temperature crack resistance of CRM are studied, and the following conclusions are drawn:The fracture work of the IDT test and the fracture energy of the SCB test can well distinguish the low-temperature performance of CRM under different gradations and emulsified asphalt contents, and the data variability is less than 15%, which is recommended as the low-temperature performance evaluation index of CRM.For fine gradation (AC-13), medium gradation (AC-20), and coarse gradation (AC-25), CRM with medium gradation (AC-20) has higher low-temperature performance. Compared with the fine gradation, the fracture work and the fracture energy are at least 32% and 24% higher, respectively, and 52% and 41% higher, respectively, compared with the coarse gradation.Polyester fiber has good dispersion, which can play a role in reinforcement consolidation and restraint of crack growth, and has a significant effect on improving the low-temperature performance of CRM. Compared with no fiber, when the fiber content is 0.3%, the fracture work and the fracture energy of CRM are increased by at least 73% and 30%, respectively.Through the comparative analysis of the low-temperature performance of CRM under different test schemes, the polyester fiber is recommended to be selected, with a content of 0.3%. In the future, the influence of composite blending of fiber and regenerant on the low-temperature performance of CRM should be further studied.

## Figures and Tables

**Figure 1 materials-15-07867-f001:**
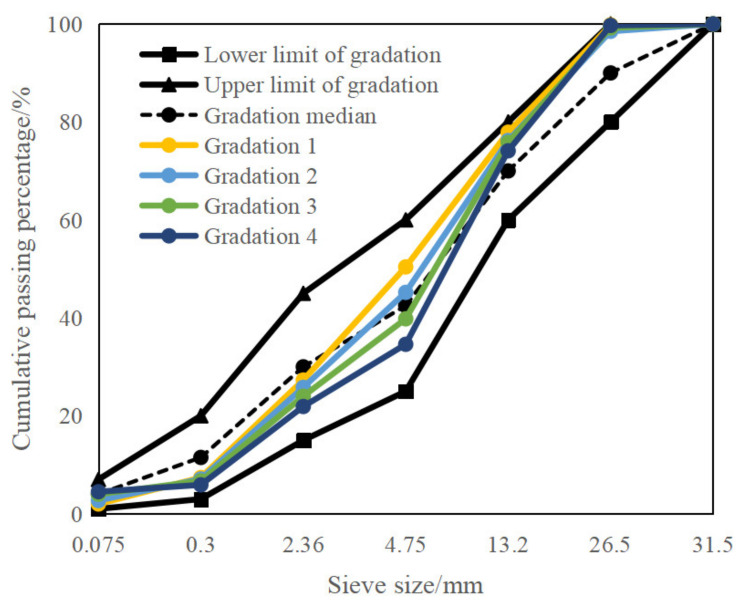
Different gradation types of CRM.

**Figure 2 materials-15-07867-f002:**
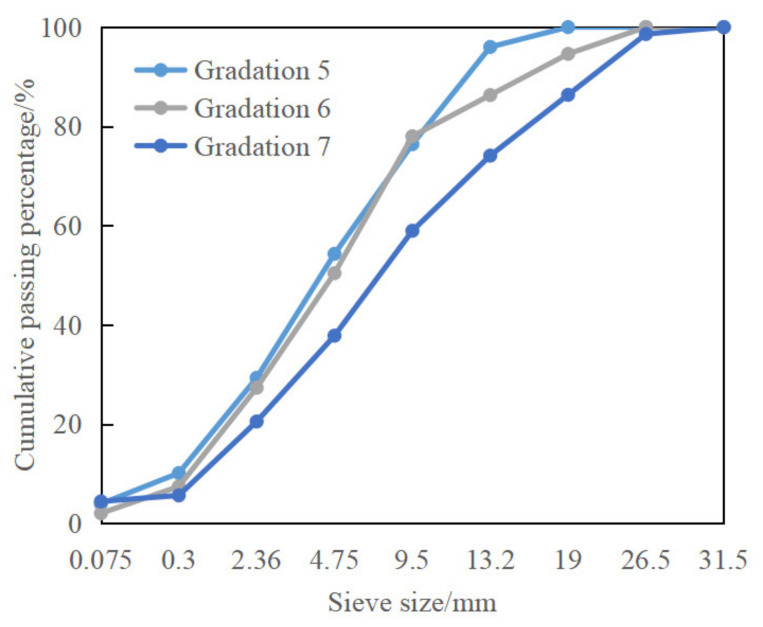
Three gradation types of CRM.

**Figure 3 materials-15-07867-f003:**
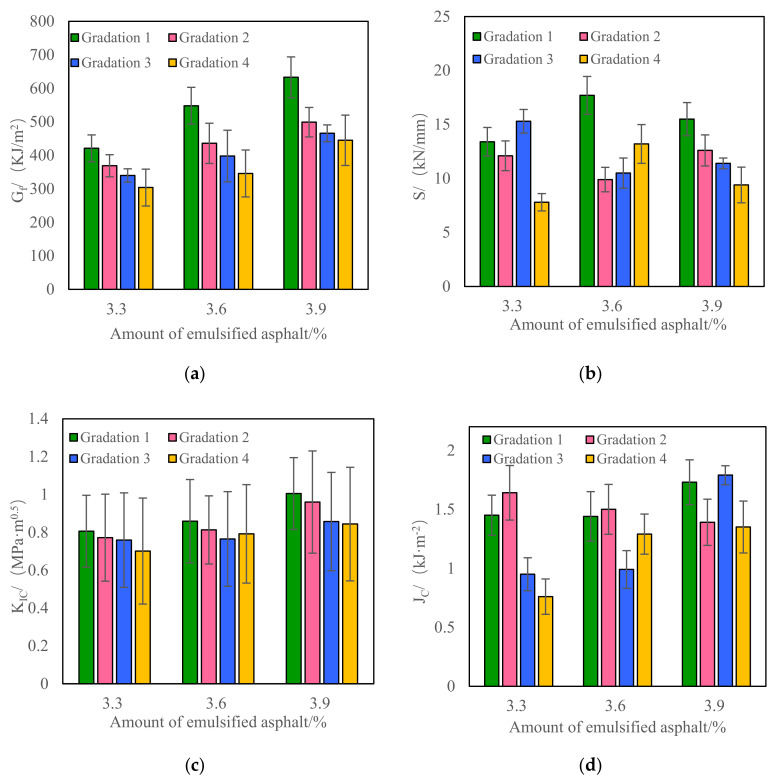
Low-temperature performance test results of CRM under different grades and emulsified asphalt dosage (SCB test). (**a**) Gf; (**b**) S; (**c**) KIC; (**d**) JC.

**Figure 4 materials-15-07867-f004:**
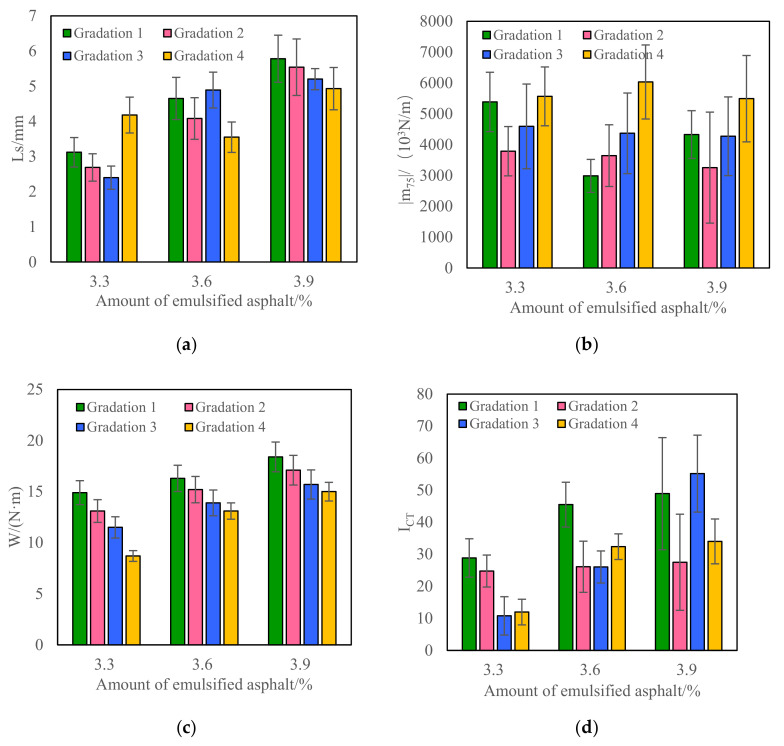
Low-temperature performance test results of CRM with different gradations and emulsified asphalt contents (IDT test). (**a**) Ls; (**b**) |m_75_|; (**c**) W; (**d**) *I*_CT_.

**Figure 5 materials-15-07867-f005:**
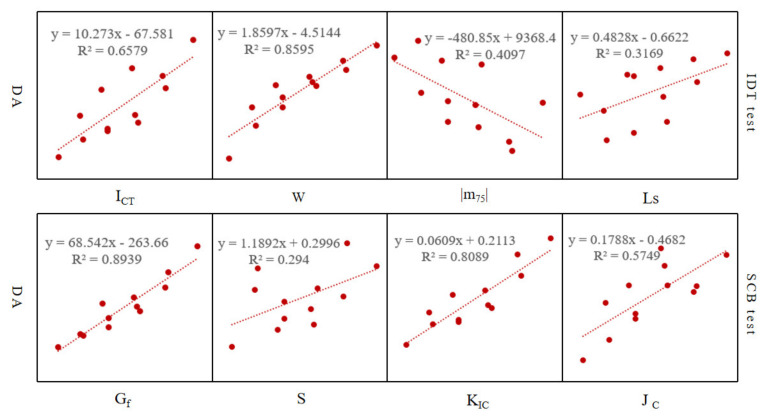
Correlation analysis between new asphalt film thickness and low-temperature crack resistance index.

**Figure 6 materials-15-07867-f006:**
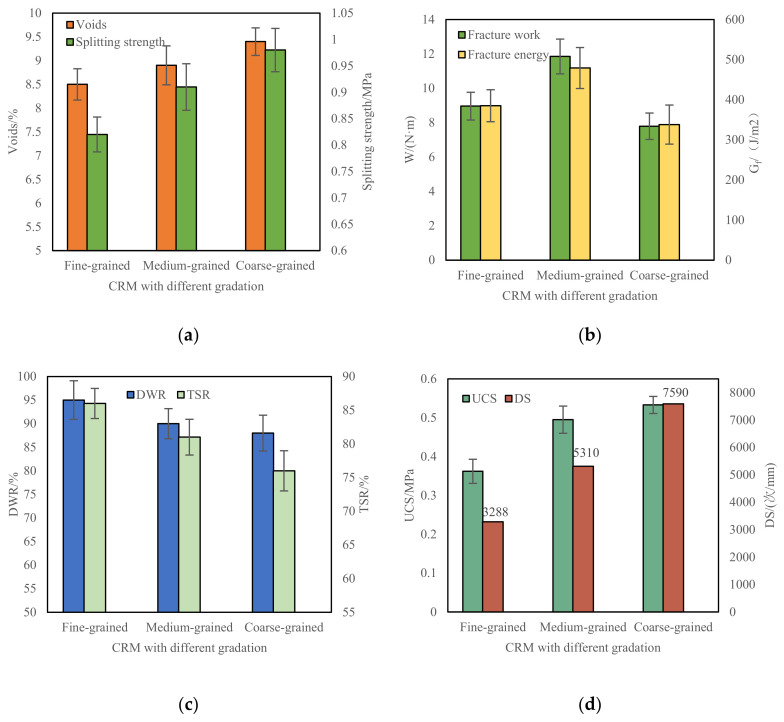
Performance and voids test results of CRM under different gradation. (**a**) Voids and R_T_; (**b**) W and G_f_; (**c**) DWR and TSR; (**d**) UCS and DS.

**Figure 7 materials-15-07867-f007:**
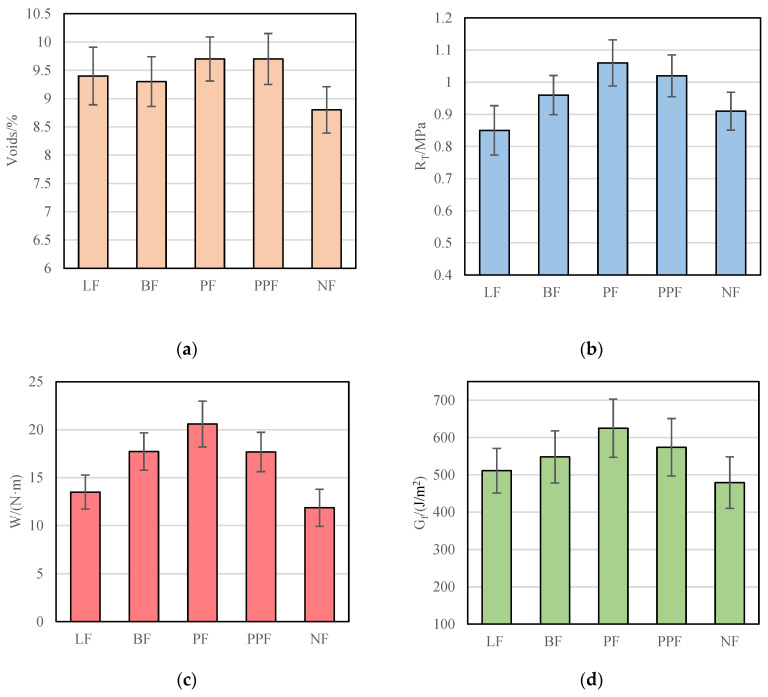
Low-temperature performance of CRM under different fiber types. (**a**) Voids; (**b**) R_T_; (**c**) W; (**d**) G_f_.

**Figure 8 materials-15-07867-f008:**
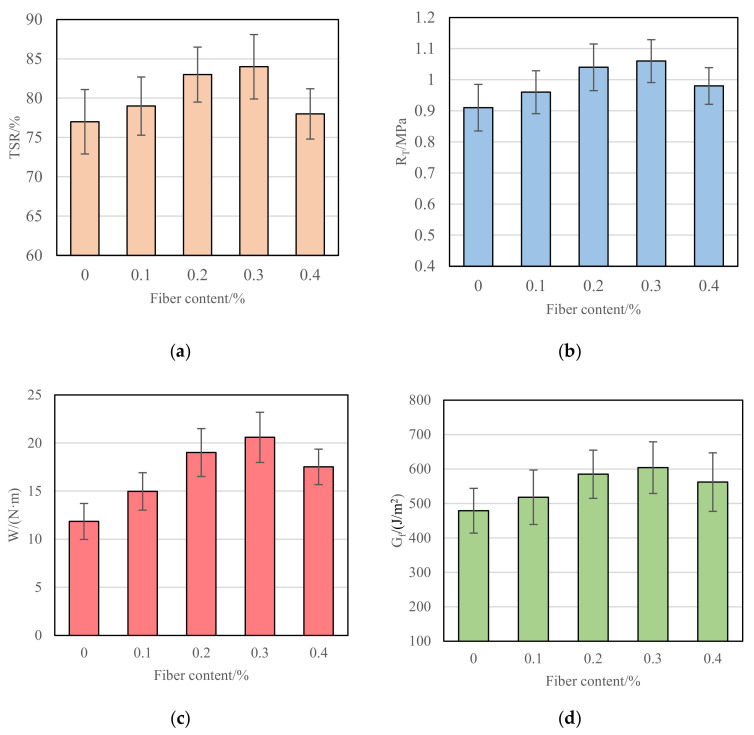
Low-temperature performance of CRM under different fiber contents. (**a**) TSR; (**b**) R_T_; (**c**) W; (**d**) G_f_.

**Table 1 materials-15-07867-t001:** Test results of emulsified asphalt properties.

Items		Units	Values	Requirement	Methods	Values	Methods
Residue on sieve (1.18 mm sieve)		%	0.03	≤0.1	T0652-1993	--	--
Particle charge		N/A	Cation (+)	Cation (+)	T0653-1993	Cation (+)	ASTM D 244
Demulsification speed		N/A	Slow crack	Slow crack	T0658-1993	Slow crack	ASTM D 244
Siebert viscosity (Vs)		s	24.98	7~100	T0621-1993	26.77	ASTM D 88
Evaporation residual component content		%	62.6	≥62	T0651-1993	63.1	ASTM D 244
Residue	Penetration (25 °C)	0.1 mm	71.1	50~300	T0604-2000	72.3	ASTM D 5
	Ductility (15 °C)	cm	56.8	≥40	T0605-1993	62	ASTM D 133
	Softening point(°C)	°C	45.8	N/A	T0606-2000	52.2	ASTM D 36
Storage stability: 1 d		%	0.8	≤1	T0655-1993	--	--
Adhesive		N/A	>2/3	≥2/3	T0654-1993	--	--

**Table 2 materials-15-07867-t002:** Technical properties of mineral powder.

Test Items	Apparent Relative Density	Water Content (%)	Hydrophilicity Coefficient	Heating Stability	CaCO_3_ Content (%)
Measured value	2.759	0.3	0.58	Good	96
Technical requirement	≥2.5	≤1.0	<1.0	Measured	≥90

**Table 3 materials-15-07867-t003:** Technical indexes of cement.

Test Items	Fineness Screen Residue (%)	Initial Setting Time (min)	Final Setting Time (min)	Compressive Strength at 3 Days (MPa)	Bending Strength at 3 Days (MPa)
Measured value	1.6	210	370	24.7	6.2
Technical requirement	≤10	≥180	≥360	≥22	≥4.0

**Table 4 materials-15-07867-t004:** Technical properties of different fiber types.

Items	Units	Different Fiber Types			
		Polyester Fiber	Polypropylene Fiber	Basalt Fiber	Lignin Fiber
Color	--	White	White	Brown	Gray
Diameter	μm	20 ± 5	18–65	16	46
Length	mm	6	6	6	1.1
Density	g/cm^3^	1.36	0.91	4.36	0.91
Melting point	°C	258	165–175	1500	220
Moisture absorption	%	2	1.1	0.1	29
Tensile strength	MPa	≥700	≥500	≥2000	<300
Elongation at break	%	30 ± 9	15–20	2.7	25

**Table 5 materials-15-07867-t005:** Passing rate of each sieve hole of different types of grading.

Gradation Types	Passing Rate of the Following Sieve Holes (mm) /%								
	31.5	26.5	19.0	13.2	9.5	4.75	2.36	0.3	0.075
Gradation 5	100	100	100	96.0	76.4	54.3	29.3	10.1	3.9
Gradation 6	100	100	94.6	/	78.0	50.4	27.3	7.4	2
Gradation 7	100	98.6	/	74.1	/	37.8	20.5	5.6	4.4

**Table 6 materials-15-07867-t006:** New asphalt film thickness of CRM under different gradations and emulsified asphalt dosages.

Different Calculation Conditions		Asphalt Film Thickness of Emulsified Asphalt CRM under the Following Gradations			
		Gradation 1	Gradation 2	Gradation 3	Gradation 4
Thickness of new asphalt film (DA)/μm	3.3%	10.7	9.7	8.9	8.1
	3.6%	11.6	10.6	9.7	8.8
	3.9%	12.5	11.5	10.5	9.5

## Data Availability

The data used to support the findings of this study are included within the article.

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
