# Peer review of "Study on Low-Temperature Index and Improvement Measures of Emulsified Asphalt Cold Recycled Mixture"

_materials, 2022, doi:10.3390/ma15217867_

Round 1

Reviewer 1 Report

This article includes a study to establish the low temperature evaluation index and to improve CRM performance. The study design is to examine CRM samples using SCB and IDT tests, which are utilized to assess their performance at low temperatures under various test schemes. The goal of this research is to determine several low temperature evaluation indices and to obtain an appropriate low temperature evaluation index for CRM. The authors claim that this evaluation index may be used to examine the low-temperature performance of CRM under various gradations, fiber types, and contents. According to my evaluation, the work meets the scientific requirements of being acceptable, rational, concise, and having a systematic and consistent sequence of discussion. However, there are several notes that authors may employ to improve the articles they write, such as: 
1. The order number 1 is written on page 2 of the chapter Raw Materials and Experimental Design. It should be written "2." Following that, adjustments must be made to additional sub-chapters. 
2. Table 1 shows the test results of emulsified asphalt performance using the Chinese standard test technique. The authors might also include international standards like ASTM or AASHTO to ensure that the testing standards used in this study are followed. 
3. On page 3, sub-chapter 2, it says RAP. The author must define the term and its abbreviation, RAP. 
4. The author employs a variety of fibers in this investigation, although it is not mentioned in the discussion. What is the purpose of this fiber? Is there a particular reason that the authors can justify? 
5. Although the authors claim that the mineral powder, cement, and water used fulfill the JTG F40-2004 criteria, I suggest that the results of the inspection of these materials, as well as the test technique and the value of the specifications utilized, should be given in tabular form. This is to demonstrate the existence of components in the CRM mixture. 
6. On page 5, the authors employ a gradation variation in the CRM mixture (Gradations 5, 6, and 7). It is preferable for the author to add a graph of the grain gradation as well as other gradations employed in this combination in order to explain the review of this study (Table 4). Is the gradation picture in Figure 1 complete with all of the gradations utilized in this study? Because only information regarding the type of gradation 1 is provided in the figure. 
7.Are there any scientific or technical reasons why the authors use 0.3% of the total types of fibers (Page 6, lines 202–203)? 
8. In response to the author's claim on page 7, lines 222–233, that stiffness and JC in CRM cannot be used to determine CRM performance at low temperatures, the author should be able to present scientific justification as well as some opinions and similar research results that support the author's claim. Because I think stiffness and Jc are still important in figuring out how resistant a mixture is to cracking and deforming. 
In Point 3 of the Conclusion, the authors could explain more about the scientific reasons for using different fibers and how that affects the value of making CRM work much better. 

Author Response

Comments and Suggestions for Authors

This article includes a study to establish the low temperature evaluation index and to improve CRM performance. The study design is to examine CRM samples using SCB and IDT tests, which are utilized to assess their performance at low temperatures under various test schemes. The goal of this research is to determine several low temperature evaluation indices and to obtain an appropriate low temperature evaluation index for CRM. The authors claim that this evaluation index may be used to examine the low-temperature performance of CRM under various gradations, fiber types, and contents. According to my evaluation, the work meets the scientific requirements of being acceptable, rational, concise, and having a systematic and consistent sequence of discussion. However, there are several notes that authors may employ to improve the articles they write, such as: 
1. The order number 1 is written on page 2 of the chapter Raw Materials and Experimental Design. It should be written "2." Following that, adjustments must be made to additional sub-chapters. 

Response:We are very sorry for the clerical error here, the chapter number has been corrected and adjusted.

  1. Table 1 shows the test results of emulsified asphalt performance using the Chinese standard test technique. The authors might also include international standards like ASTM or AASHTO to ensure that the testing standards used in this study are followed. 

Response:According to the reviewer’s comments, the ASTM standard test results of emulsified asphalt performance are added. The added part has been marked in red in the text.

  1. On page 3, sub-chapter 2, it says RAP. The author must define the term and its abbreviation, RAP. 

Response:Thank you for this comment. According to the reviewer’s comments, the explanation of RAP is supplemented in the paper.

  1. The author employs a variety of fibers in this investigation, although it is not mentioned in the discussion. What is the purpose of this fiber? Is there a particular reason that the authors can justify? 

Response:As previously mentioned, fiber in asphalt mixture has the function of reinforcing support, dispersing and transferring stress, and limiting the relative slip of mineral aggregate. It has achieved good results in slowing down the fatigue crack growth of mixture, and can be used to improve the low-temperature crack resistance of CRM . Therefore, we employ a variety of fibers in this investigation. References are as follows:

Morea, F., & Zerbino, R. (2018). Improvement of asphalt mixture performance with glass macro-fibers. Construction and Building Materials, 164, 113-120.

Luo, D., Khater, A., Yue, Y., Abdelsalam, M., Zhang, Z., Li, Y., ... & Iseley, D. T. (2019). The performance of asphalt mixtures modified with lignin fiber and glass fiber: A review. Construction and Building Materials, 209, 377-387.

  1. Although the authors claim that the mineral powder, cement, and water used fulfill the JTG F40-2004 criteria, I suggest that the results of the inspection of these materials, as well as the test technique and the value of the specifications utilized, should be given in tabular form. This is to demonstrate the existence of components in the CRM mixture. 

Response:According to the reviewer’s comments, test results , test technique , the value of the specifications utilized of cement and mineral powder are supplemented, as shown in Table 2 and Table 3.

  1. On page 5, the authors employ a gradation variation in the CRM mixture (Gradations 5, 6, and 7). It is preferable for the author to add a graph of the grain gradation as well as other gradations employed in this combination in order to explain the review of this study (Table 4). Is the gradation picture in Figure 1 complete with all of the gradations utilized in this study? Because only information regarding the type of gradation 1 is provided in the figure. 

Response:Thank you for this comment. According to the reviewer’s comments, the curves of gradations 5, 6, and 7 are added to better represent the changes of different gradations.

We are very sorry that there may be a problem with the presentation of the Figure 1. The Figure 1 has been readjusted and the further information on gradations 2, 3, & 4 has been provided.

7.Are there any scientific or technical reasons why the authors use 0.3% of the total types of fibers (Page 6, lines 202–203)? 

Response:For the selection of 0.3% fiber content, the authors refer to the previous research results of Jiang, Cai and others, and "0.3% fiber content test is recommended". Simultaneously, it will avoid that each fiber is tested with different content, and the test cost and time will be saved. References are as follows:

Liu H., Jiang Y., Hu Y., Ye W., et al. (2018) The influence of gradation on the strength of cold recycled asphalt emulsion mixture[J]. Journal of Building Materials (03), 503-510. 

Cai Y. (2018) Study on the Performance and Design Method of Cold Recycled Asphalt Emulsion Mixture Formed by Vertical Vibration[D]. Xi'an: Doctoral Dissertation of Chang'an University. (in Chinese)

  1. In response to the author's claim on page 7, lines 222–233, that stiffness and JC in CRM cannot be used to determine CRM performance at low temperatures, the author should be able to present scientific justification as well as some opinions and similar research results that support the author's claim. Because I think stiffness and Jc are still important in figuring out how resistant a mixture is to cracking and deforming. 
    In Point 3 of the Conclusion, the authors could explain more about the scientific reasons for using different fibers and how that affects the value of making CRM work much better. 

Response:Thank you for this comment. Before exploring the low-temperature evaluation index of CRM, we also learned that previous studies believed that stiffness and JC had certain evaluation effects, but our test results were really not ideal. It may be that stiffness and JC were originally used to evaluate the crack resistance of hot mix asphalt mixture, while cement was added to the CRM, which belongs to semi flexible materials. Due to the change of material properties, stiffness and JC are no longer applicable. Later, we will continue to supplement test samples to continuously verify the applicability of stiffness and JC.

According to the reviewer’s comments, in Point 3 of the Conclusion has been reorganized. The revised part has been marked in red in the text.

Reviewer 2 Report

Overall, the manuscript presented an interesting study. However, here are some comments/suggestions to further improve the paper:

1. Further information on gradations 2, 3, & 4 should be provided. It would help reader to better understand the discussion of result and the impact of aggregate grading on the test results.

2. To use abbreviation/ short form, write the full name in the first instance and follow immediately by the abbreviated or designated version in bracket. Avoid specifying both full name and abbreviation/short form repeatedly.

3. For the citation or reference, when mentioning the author names, avoid using full name, it can be done by the author last name and should follow the journal format guidelines

4. Referring to Table 1, please check the symbol used

5. In table 1, referring to softening point test, what is that mean by 5 deg C

6. In Fig. 1, the figure legend is not complete. Indicators for gradings 2-4 are missing.

7. Referring to unit used for the fracture energy "J/m2", the number "2" should be superscripted, please check others as well

Author Response

Comments and Suggestions for Authors

Overall, the manuscript presented an interesting study. However, here are some comments/suggestions to further improve the paper:

  1. Further information on gradations 2, 3, & 4 should be provided. It would help reader to better understand the discussion of result and the impact of aggregate grading on the test results.

Response:We are very sorry that there may be a problem with the presentation of the picture. The picture has been readjusted and the further information on gradations 2, 3, & 4 has been provided.

  1. To use abbreviation/ short form, write the full name in the first instance and follow immediately by the abbreviated or designated version in bracket. Avoid specifying both full name and abbreviation/short form repeatedly.

Response:Thank you for this comment. According to the reviewer’s comments, The authors carefully checked the full text, and the abbreviations have been expressed in the correct form.

  1. For the citation or reference, when mentioning the author names, avoid using full name, it can be done by the author last name and should follow the journal format guidelines

Response:Thank you for this comment. According to the reviewer’s comments, The authors have revised the incorrect citation format in the introduction and references, which has been marked in red.

  1. Referring to Table 1, please check the symbol used

Response:The authors carefully checked Table 1 and corrected inappropriate symbols.

  1. In table 1, referring to softening point test, what is that mean by 5 deg C

Response:We are very sorry for the clerical error here, which has been corrected in Table 1 and marked in red.

  1. In Fig. 1, the figure legend is not complete. Indicators for gradings 2-4 are missing.

Response:The Fig. 1 has been readjusted and the further information on gradations 2, 3, & 4 has been provided.

  1. Referring to unit used for the fracture energy "J/m2", the number "2" should be superscripted, please check others as well

Response:Thank you for this comment. The authors carefully checked the full text, and the units of writing errors have been corrected.

Reviewer 3 Report

paper on" Study on Low-Temperature Index and Improvement Measures of Emulsified Asphalt Cold Recycled Mixture" is good topic and under the journal scope. However, minor revision required as find below:

1- The introduction must be stronger in terms of addressing the Reseach gap. Adding strong statement of research gap and linked it to the objectives. This is must to show the study's novelty. 

2- Results and discussion: to make the paper sounds more effective, authors must add strong argument and must compare the current results with findings from old/previous studies. [ This must be for each data tests].  

finally, add a new point to the conclusion to show the ideal contents and mixtures performance. And any future recommendation!!

Author Response

Comments and Suggestions for Authors

paper on" Study on Low-Temperature Index and Improvement Measures of Emulsified Asphalt Cold Recycled Mixture" is good topic and under the journal scope. However, minor revision required as find below:

1- The introduction must be stronger in terms of addressing the Reseach gap. Adding strong statement of research gap and linked it to the objectives. This is must to show the study's novelty. 

Response ï¼šThank you for this comment. According to the reviewer’s comments, the authors strengthen the statement of research gaps and link them to the objectives of this paper. The revised part has been marked in red in the text.

2- Results and discussion: to make the paper sounds more effective, authors must add strong argument and must compare the current results with findings from old/previous studies. [ This must be for each data tests].  

finally, add a new point to the conclusion to show the ideal contents and mixtures performance. And any future recommendation!!

 Response ï¼šAccording to the reviewer’s comments, the authors have demonstrated the reliability of the test results by referring to previous studies. And a new conclusion is added, including suggestions for future research. The revised part has been marked in red in the text.